# Effects of COVID-19 Home Confinement on Behavior, Perception of Threat, Stress and Training Patterns of Olympic and Paralympic Athletes

**DOI:** 10.3390/ijerph182312780

**Published:** 2021-12-03

**Authors:** María José Martínez-Patiño, Francisco Javier Blas Lopez, Michel Dubois, Eric Vilain, Juan Pedro Fuentes-García

**Affiliations:** 1Faculty of Sciences of Education and Sport, University of Vigo, 36005 Vigo, Spain; mjpatino@uvigo.es (M.J.M.-P.); javier.blas@icloud.com (F.J.B.L.); 2Groupe d’Étude des Méthodes de l’Analyse Sociologique de la Sorbonne (GEMASS), CNRS, Sorbonne Université, 75006 Paris, France; Michel.DUBOIS@cnrs.fr; 3Centre National de la Recherche Scientifique (Cnrs), International Research Laboratory “Epidapo” (Epigenetics, Data, Politics), Washington, DC 20052, USA; 4Center for Genetic Medicine Research, Children’s National Hospital, Washington, DC 20010, USA; 5Department of Genomics and Precision Medicine, George Washington University, Washington, DC 20052, USA; 6Faculty of Sport Science, University of Extremadura, 10003 Cáceres, Spain; jpfuent@unex.es

**Keywords:** lockdown, harmful behavioral, elite athletes, sport adapted, psychological variables

## Abstract

Background: The aims of this study were to analyze the effects of the COVID-19 pandemic and its subsequent confinement on behaviors, perception of threat, stress, state of mind and training patterns among Olympic and Paralympic level athletes. Methods: Data gathering was performed utilizing an online questionnaire during imposed confinement. A correlational design with incidental sampling for convenience was used. All the variables were analyzed by age, gender, academic training, type of participation and sport specialty on a population composed of 447 Olympic (age: 26.0 ± 7.5 years) and 64 Paralympic (age: 28.4 ± 10.5 years) athletes. Results: The athletes trained more than twice as many hours before than during confinement. Most of the athletes recognized that their best athletic performance diminished due to the COVID-19 confinement but that will recover after the pandemic and its confinements. Almost half of the athletes declared they were more tired than normal and had difficulty sleeping, while more than half ate more or less as usual. Paralympic athletes reported they felt more capable to cope with personal problems and life events and felt less lonely during the confinement than the Olympians. The athletes from team sports reported to be more affected in their training routine than athletes of individual sports, seeing their athletic performance more affected. Athletes in individual sports felt more able to cope with personal problems than athletes in team sports. Female athletes were significantly more tired and reported more difficulty sleeping than male athletes. Conclusion: The situation caused by COVID-19 has had significant effects on the behavior, perception of threat, stress and training patterns of Olympic and Paralympic athletes preparing for the 2020 Tokyo Olympics. It is necessary that sports institutions reinforce mechanisms of help for athletes during future situations of confinement.

## 1. Introduction

In December 2019, a novel coronavirus (Severe Acute Respiratory Syndrome Coronavirus-2 (SARS-CoV-2) [1]) was reported in China and eventually triggered an international public health emergency. On 11 March 2020, this coronavirus disease was declared a pandemic by the World Health Organization (WHO) [2], representing the most serious respiratory virus since the 1918 H1N1 influenza pandemic [3].

On 14 March 2020, in the context of multiple global confinement orders, the Spanish government declared a nationwide lockdown, ordering people to stay at home [4] and approving the last extension of the “state of alarm” until 21 June 2020 [5]. Our study lasted from 2 to 22 September. To place the investigation in a global perspective, halfway through it, on 13 September 2020, there were 28,637,952 confirmed cases and 917,417 deaths worldwide [2]. In Spain, there were 12,113 confirmed cases and 636 deaths per million people. 

During the current global home confinement situation due to the COVID-19 pandemic, many individuals were exposed to an unprecedented stressful situation of unknown duration that was a source of anxiety, fear and depression or disrupted sleep due to the negative appraisal of the situation and self-protection behaviors [6,7]. Here, we are focusing on a unique population of elite athletes to assess their response to the extraordinary situation of a pandemic lockdown.

In the context of COVID-19, the Olympic and Paralympic Games may be a major source of insecurity to many worldwide athletes and lead to changes to everyday lives and potentially prolonged psychological distress [8], with their potential postponement being an additional stressor. The threat of COVID-19 may cause the athlete to lose concentration, motivation and the desire to continue preparing for the Olympics with the same energy as they did up until then [9,10]. Information about the perception of threat would be fundamental to understand their stress responses and to define professional interventions for these athletes so that they continue to prepare for the Olympiad optimally.

The confinement situation caused by COVID-19 has had significant effects on the perception of the personal and professional threat of Olympic and Paralympic athletes preparing for the 2020 Tokyo Olympics [11]. A study with Olympic and Paralympic athletes during confinement showed how they perceived how confinement negatively affects their workouts, even if it does not affect their performance, or approval of the suspension of the Tokyo Olympics [10]. Neuroticism and psychological inflexibility correlated with the greatest negative feelings and the perception that quarantine would negatively affect their sports performance. Faced with this same confinement situation generated by COVID-19, a study on a sample of 450 chess players showed that long training durations, tournaments and games could provide chess players a certain advantage in preparing for quarantine situations [12]. 

The aim of the present research is to produce new knowledge on the response of elite level athletes—a largely understudied population—to COVID-19 confinement. Our aim is not only to account for the effect of the COVID-19 pandemic and its subsequent confinement on Olympic- and Paralympic-level athletes’ perceptions, state of mind and training patterns, but also to assess the different responses of high-performance athletes to the pandemic based on a wide set of variables such as gender, academic training and sport discipline.

## 2. Methods

In this study, data gathering was performed utilizing an online questionnaire sent to participants via the online platform SurveyMonkey (https://www.surveymonkey.com/ (accessed on 2 September 2020)). A correlational design with incidental sampling for convenience was used. During imposed lockdown/confinement, this methodology was the most appropriate for obtaining data.

### 2.1. Participants

The inclusion criteria used for the present research was that athletes were selectable by their corresponding sports federation with options to participate in the 2020 Tokyo Olympic or Paralympic Games. The total population studied included 511 Olympic- and Paralympic-level athletes (nationally ranked or in the process of participating in the Tokyo Olympic Games); 447 Olympic athletes (Median age = 25.97 years; SD = 7.46) and 64 Paralympic athletes (Median age = 28.44 years; SD = 10.50). The median age of the total population was 26.28 years (SD = 7.94), and gender distribution was almost equally balanced: 257 males and 254 females. The detailed distribution of the population by discipline is available in Appendix A. 

### 2.2. Procedure

Due to the COVID-19 crisis and the limitation of free movement, an intentional opinion-type sampling type was used. The call to participate in the study was made through a link sent by the federations of the different sports [13]. The questionnaire was sent to 871 athletes: 678 Olympic athletes, with 447 (66%) answering, and 193 Paralympic athletes, with 64 (34%) answering. This study was completely voluntary, and no personal data through which the participants could be identified were requested. Data collection lasted for 21 days (from 2 to 22 September 2020). Before participation, experimental procedures were explained to all the participants who gave their voluntary written informed consent in accordance with the Declaration of Helsinki. All the procedures were approved by the Commission of Bioethics and Biosecurity of the University of Extremadura (Spain) (approval number: 57/2020).

Firstly, the athletes provided informed consent and then completed the following items, which we present here briefly and in full in Appendix B: Personal Information, Questionnaire on Perception of Threat from COVID-19 [14], Perceived Stress by Psychometric Properties of a European Spanish Version of the Perceived Stress Scale (PSS-10) [15], academic and training patterns information and individual information, and perceptions and harmful behavioral about the COVID-19 crisis.

### 2.3. Statistical Analysis

The data (N = 511), grouped according to age considering M = 26.28 years (N_25orLess_ = 286 (56%), N_26orMore_ = 225 (44%)), gender (N_male_ = 257 (50.3%), N_Female_ = 254 (49.7%)), academic training (N_high school_ = 168 (32.9%), N_Professional_ = 96 (18.8%), N_University_ = 247 (48.3%)), Olympic and Paralympic participation (N_Olympic_ = 447 (87.5%), N_Paralympic_ = 64 (12.5%)), sport specialty (N_Individual_ = 328 (64.2%), N_Collective_ = 183 (35.8%)), ranked or with high options to participate in the Tokyo 2020 (N_Yes_Tokyo_ = 261 (51.1%), N_No_Tokyo_ = 250 (48.9%)). 

Before the data analyses, the normal distribution of the dependent variables was tested with the Kolmogorov–Smirnov test. All test results were statistically significant (*p* < 0.05). Consequently, nonparametric tests were used. Wilcoxon signed-rank test to compare two related samples was used. Possible differences in dependent variables between groups were tested with the nonparametric Mann–Whitney U test with the Bonferroni correction, which is recommended for smaller samples and might be less sensitive to sample size differences between the groups [16]. Effect size was calculated and interpreted using the following thresholds: small (|r| = 0.1), medium (|r| = 0.3), large (|r| = 0.5) and very large (|r| = 0.7) [17].

In case of three groups (i.e., academic training groups), the Kruskal–Wallis test, also known as the analysis of variance on ranks, was used, and the epsilon-squared estimate of effect size was subsequently calculated [18] using the following interpretation for r: negligible (|E_R_^2^| ≤ 0.01), weak (|E_R_^2^| ≤ 0.04), moderate (|E_R_^2^| ≤ 0.16), relatively strong (|E_R_^2^| ≤ 0.36), strong (|E_R_^2^| ≤ 0.64) and very strong (|*r*| ≤ 1) [19]. 

The chi-square tests were performed (X^2^) to analyze the ordinal categorical variables of the individual, sport and perceptions information about the COVID-19 crisis. 

Additionally, for psychological profile and perceived stress variables, a bivariate correlation analysis between all these variables was performed using Spearman’s correlation thresholds correlation.

## 3. Results

Table 1 showed the descriptive data, such as the number of subjects (N), mean (M) and standard deviation (SD) of the age, anthropometric dimensions, training patterns, individual perceptions about COVID-19 crisis and psychological profile.

### 3.1. Hours of Training per Week

When comparing the number of hours per week that the athletes trained before and during the confinement decreed by the COVID-19, it is observed that the athletes trained more than twice as many hours before (M = 17.30) than during confinement (M = 8.44) (Z = −17.291, *p* < 0.001, effect size (r) = 0.78, very large).

### 3.2. Age

Statistical analysis revealed that the group under 26 years of age felt they had less control of all aspects of life in the last month than the group aged 26 and over (Z = 2.002, *p* < 0.045, effect size (r) = 0.09, small); athletes under 26 years of age most frequently felt angry/upset due to events that occurred to them that were beyond their control (Z = 2.020, *p* < 0.043, effect size (r) = 0.09, small) and had a greater feeling that problems pile up that cannot be overcome than the group aged 26 and over (Z = 1.988, *p* < 0.047, effect size (r) = 0.09, small). Finally, the group aged 26 and over was more concerned about their financial situation during the confinement than the group aged 25 and under (Z = 3.909, *p* < 0.001, effect size (r) = 0.17, small), while, inversely, the group aged 25 and under longed to be able to interact with other athletes during the confinement more (Z = 3.282, *p* < 0.001, effect size (r) = 0.15, small). On the other hand, the chi-square test showed significant differences between the two age groups in relation to whether the athletes approved of the confinement imposed in their country (X^2^ = 6.48, *p* < 0.039): the group under 26 years of age (Yes = 170, 60.3%; No = 36, 12.7%; Neutral = 76, 27%) and the group aged 26 and over (Yes = 146, 65.2%; No = 38, 17%; Neutral = 40, 17.8%).

### 3.3. Gender

Statistical analysis revealed that the female compared to the male group was more emotionally affected by COVID-19 (e.g., angry, fearful, upset or depressed) (Z = 2.571, *p* < 0.010, effect size [r] = 0.11, small). Women also perceived that they had been more affected than men by an unexpected event in the last month (Z = 2.668, *p* < 0.008, effect size [r] = 0.12, small), felt they had less control over important events in their life (Z = 4.966, *p* < 0.001, effect size [r] = 0.22, small), more nervous/anxious or stressed (Z = 5.099, *p* < 0.001, effect size [r] = 0.23, small), less able to cope with their personal problems (Z = 2.050, *p* < 0.040, effect size [r] = 0.09, small), less able to face their responsibilities (Z = 2.728, *p* < 0.006, effect size [r] = 0.12, small), less in control of all aspects of their life (Z = 3.681, *p* < 0.001, effect size [r] = 0.16, small), more angry/upset due to events that occurred that were out of their control (Z = 4.442, *p* < 0.001, effect size [r] = 0.20, small), greater feeling that problems pile up that cannot be overcome (Z = 2.941, *p* < 0.003, effect size [r] = 0.13, small). On the other hand, the training routines of women were disrupted more by COVID-19 than those of men (Z = 2.279, *p* < 0.023, effect size [r] = 0.10, small), with women also feeling more stressed (pressured, tense or overwhelmed) than usual because of the confinement (Z = 4.037, *p* < 0.001, effect size [r] = 0.18, small), more worried about reducing their athletic capability because of the confinement (Z = 3.074, *p* < 0.002, effect size [r] = 0.14, small) and more lonely than men (Z = 2.333, *p* < 0.020, effect size [r] = 0.10, small). 

### 3.4. Academic Training

The results showed that athletes in the high school group evaluated the gravity of the COVID-19 pandemic as less serious than the athletes in the professional training and university training groups (*p* < 0.011, effect size [E_R_^2^] = 0.018, weak). On the other hand, the athletes in the professional training group saw their athletic performance as more affected by the confinement than the high school and university training groups (*p* < 0.001, effect size [E_R_^2^] = 0.027, weak), showing the group of athletes with professional training to be more worried about reducing their athletic capability because of the confinement than the group with university training (*p* < 0.021, effect size [E_R_^2^] = 0.015, weak), the group with university training to be more worried about their financial situation than the high school group (*p* < 0.025, effect size [E_R_^2^] = 0.015, weak) and, conversely, the high school missed interacting with other athletes more than the group with university training (*p* < 0.020, effect size [E_R_^2^] = 0.016, weak).

### 3.5. Olympic and Paralympic Participation

Statistical analysis revealed that in the previous month, the Paralympic compared to the Olympic athletes felt more able to cope with their personal problems (Z = 2.593, *p* < 0.010, effect size [r] = 0.12, small), felt that life events are going well more frequently (Z = 1.989, *p* < 0.047, effect size [r] = 0.09, small) and felt less lonely than Olympic athletes (Z = 2.744, *p* < 0.006, effect size [r] = 0.12, small). On the other hand, the chi-square test showed significant differences between the two types of participation groups in relation to whether the athletes approved of the confinement imposed in their country (X^2^ = 10.13, *p* < 0.006): Olympic group (Yes = 265, 59.9%; No = 71, 16.1%; Neutral = 106, 24%) and Paralympic group (Yes = 51, 79.7%; No = 3, 4.7%; Neutral = 10, 15.6%).

### 3.6. Sport Specialty

The results showed that in the last month, the athletes in individual sports felt more capable of coping with personal problems than athletes in collective sports (Z = 2.596, *p* < 0.009, effect size [r] = 0.12, small). On the other hand, athletes of collective sports saw their training routine as disrupted by COVID-19 more than athletes of individual sports (Z = 2.716, *p* < 0.007, effect size [r] = 0.12, small), saw their athletic performance as affected by the confinement more than individual sports athletes (Z = 1.983, *p* < 0.047, effect size [r] = 0.08, small) and were also more concerned about not being able to participate in sports competitions because of the confinement (Z = 2.012, *p* < 0.044, effect size [r] = 0.09, small). On the other hand, the chi-square test showed significant differences between the two types of sport specialty groups in relation to whether the athletes approved of the confinement imposed in their country (X^2^ = 6.10, *p* <0.047): individual sport group (Yes = 192, 59.1%; No = 56, 17.2%; Neutral = 77, 23.7%) and collective sport group (Yes = 124, 68.6%; No = 18, 9.9%; Neutral = 21, 5%). Likewise, the chi-square test showed significant differences between these groups in relation to whether their food intake/eating changed during the confinement (X2 = 8.69, *p* <0.013): individual sport group (Eat More than usual = 111, 34.6%; Eat Less than usual = 86, 26.8%; Eat the same as before, no change = 124, 38.6%) and collective sport group (Eat More than usual = 48, 26.5%; Eat Less than usual = 71, 39.2%; Eat the same as before, no change = 62, 34.3%). 

### 3.7. Ranked or with High Options to Participate in the Tokyo 2020

No significant differences were observed in any of the variables analyzed between those athletes ranked or with high options to participate in Tokyo 2020 and those who had lesser options.

### 3.8. Dichotomous Questions

For the dichotomous questions (answers Yes or No) about the individual, training patterns and perceptions of the information about the COVID-19 crisis, we present the results in which there have been significant differences when comparing between the groups’ age (25 or less, or 26 or over), gender (male or female), academic training (high school, professional training or university training), participation (Olympic and Paralympic), sport specialty (individual or collective) and options to participate in Tokyo 2020 (ranked or with high options, or without serious options—low options) (Table 2).

### 3.9. Correlational Analysis

Finally, a correlational analysis of the perception of threat, stress and training patterns variables about the COVID-19 crisis is shown in Table 3.

Correlation analysis shows how the five items (1 to 5: Impact of the pandemic on oneself, estimation of the duration of the pandemic, symptoms of COVID-19, concern about the pandemic and emotional affectation by the pandemic) of the Questionnaire on Perception of Threat from COVID-19 correlate with each other, in most cases strongly, except for the one that deals with the extent to which the athlete feels symptoms due to a coronavirus infection (1), which correlates with the other four variables but not with the one that deals with how much the impact of the pandemic affects the athlete (1). On the other hand, all the items (1 to 5) of the Questionnaire on Perception of Threat from COVID-19 (6 to 15) correlate, strongly in practically all cases, positively with five of the ten items of the Perceived Stress by Psychometric Properties of a European Spanish Version of the Perceived Stress Scale (PSS) (6, 7, 8, 11, 14 and 15: Affected by an unexpected event, lack of control over important events, nervous/anxious or stressed, incapable of facing their responsibilities, angry by events that were out of control and feel the problems piling up).

The rest of the items (9, 10, 12 and 13: Able to cope with personal problems, feel that the events are going well, able to manage their problems and feel in control of all aspects) of the Questionnaire on Perception of Threat from COVID -19 correlate negatively, strongly in most cases, with each other and also with the five items of the Questionnaire on Perception of Threat from COVID-19 (1 to 5). There is a positive correlation between the athlete’s ability to control the problems in their lives (12) and the feeling of controlling life’s difficulties (13) with the ability of the athlete to cope with personal problems (9), with the frequency with which the athlete has felt that things are going well (10) and with the inability of the athlete to face responsibilities (11).

## 4. Discussion

Elite athletes have had to adapt to extraordinary circumstances during the COVID-19 lockdown. There is existing empirical evidence that adults are at high risk of depression and anxiety related to the psychosocial effects of the COVID-19 pandemic [20]. Our investigation analyzes the special circumstances of the unique population of Olympians and Paralympians.

### 4.1. Attitude toward Olympics

Although the Olympic games are widely considered to be the most important sport event for athletes, and many athletes depend on this event for their sports scholarships and sponsorships, there was high agreement on the suspension of the Tokyo Olympics and the confinement of high-performance athletes (62.5% in favor, 14.6% against and 22.9% neutral), results that are in the same line as a study also with Olympic and Paralympic athletes [10].Although the majority of athletes recognize that their best athletic performance decreased due to the COVID-19 confinement (78.2% Yes and 21.8% No), with athletes under 26 years of age more negative than the group of athletes aged 26 and over, on the other hand, the majority of athletes also believe that their athletic performance will recover after the pandemic (93.6% Yes and 6.4% No). 

### 4.2. Differences between Individual and Collective Sports Athletes

The training routine of athletes of collective sports was disrupted by COVID-19 more significantly than for those of individual sports, with their athletic performance more affected by the confinement than in individual sports, probably due to the added difficulty of having to meet with companions in order to train. Remarkably, an investigation on the repercussions of the COVID-19 pandemic on athletes through a cross-sectional study showed that athletes who had a teammate who tested positive were 15-fold more likely to be tested for COVID-19 [21].

### 4.3. Effect of Lockdown on Athletes’ Behavior

During confinement, 48.7% of the athletes declared they were more tired than normal, 47.9% had difficulty sleeping, 62.9% ate more (31.7%) or less (31.3%) than usual, 4% increased their tobacco consumption, 9.8% increased their alcohol consumption and 2% took psychotropic medications for their mental health. These results are in line with other studies in which it has been shown that most individuals exposed to an unprecedented stressful situation of unknown duration experience anxiety, fear, depression and also disrupted sleep due to a negative appraisal of the situation and self-protection behaviors [6,7]. It would be important to teach athletes to manage sleep problems as best as possible during home confinement to limit stress. Interestingly the results of our study show that female athletes were significantly more tired than men, with more difficulty sleeping and more tobacco consumption. This is in line with a study that evaluated sleep quality, insomnia and depression symptoms, perceived stress and anxiety during the COVID-19 lockdown, showing that the female gender is a parameter correlated with a worsened condition for all the examined dimensions. However, at the follow-up, women reported a reduction in the severity of insomnia and depression symptoms, perceived stress and anxiety, while male participants showed a worsening of sleep quality, insomnia symptoms and perceived stress. Thus, women seemed to have greater long-term resilience during the lockdown than men [22].

### 4.4. Effect of Education Level

The high school educated group evaluated the gravity of the COVID-19 pandemic as less serious than the athletes in the groups with professional training and university training. The group with university training was more worried about their financial situation than the group with high school education. The reasons for these differences are likely multifactorial. 

### 4.5. Effect of Sports Categories 

The Paralympic athletes felt more able to cope with personal problems, felt more frequently that life events were going well and felt less lonely during the confinement compared to Olympic athletes. This is consistent with the notion that disability, in general, and sport for people with disabilities, in particular, prepares people to better cope with adversity [23]. 

### 4.6. Effect of Sports Specialty

Something comparable to the sports categories could happen regarding the sport specialty, as the athletes in individual sports felt more capable of coping with personal problems than athletes in collective sports, consistent with studies showing that the relationship between coping strategies and precompetitive anxiety is different between individual sport (swimming) and collective sport (water polo) [24], and in this sense, individual sports athletes may be more prepared to face confinement situations.

### 4.7. Role of Sports Authorities and Coaching

Faced with COVID-19 related stressors, 50.3% of the athletes reported that they received guidance from the Sports Federation, with women perceiving this help to a significantly greater extent than men. In part, these results are consistent with our results showing that female athletes felt more emotionally affected by COVID-19, with less control over important events in their life, more nervous/anxious or stressed, less able to cope with their personal problems, less able to face their responsibilities, less in control of all aspects of their life, more angry/upset due to out-of-control events, and a greater feeling that problems pile up compared, on average, to male athletes. Women saw their training routine disrupted by COVID-19 more than men, felt more stressed (pressured, tense or overwhelmed) than usual because of the confinement, were more worried about a potential reduction in their athletic capability because of the confinement and lonelier during the confinement than men. These results are consistent with data prior to the COVID crisis, with women being twice as likely than men to experience negative emotions in stressful situations, these emotions being influenced during confinement by their perception of the world, response to uncertainty, emotional intelligence and the suitability of their home for confinement [25]. These results are also consistent with those of a study with chess players, whose results showed that women reported a higher level of behaviors to avoid infection than men [12].

Finally, in terms of help from the coach, 61.8% of athletes perceived that they received help from their coach, with this help being perceived to a greater extent by athletes 25 years of age or younger than those of 26 years of age or older. Studies [26] have shown that people who tended to report difficulties in regulating emotion are younger [26], with more maladaptive, reckless and careless behaviors. Thus, it is necessary to reinforce and target more specifically the mechanisms of help and advice from sports institutions to athletes in the face of the pandemic and other stressors.

### 4.8. Correlations

The results of the correlational analysis of perception of threat, stress and training patterns variables about the COVID-19 crisis show, in most cases, strong correlations between the five items of the Questionnaire on Perception of Threat from COVID-19 and among the ten items of the Perceived Stress by Psychometric Properties of a European Spanish Version of the Perceived Stress Scale (PSS). Likewise, important correlations are shown between items from both questionnaires. 

Thus, although some athletes have successfully adapted to the demands of the COVID-19 crisis, many have experienced difficulties adjusting, showing psychological complications including increased stress, anxiety and depression, as seen in prior studies [27].

This shows the importance of psychological training in high performance athletes, since sports persons who effectively use the available cognitive resources and successfully implement coping strategies will be best equipped to manage mental distress. Different strategies for athletes in isolation due to COVID-19 can be proposed, such as developing an increased awareness of Mental Toughness as a resistance resource that protects against stress, which can improve the ability to cope with COVID-19-related challenges [28]. 

## 5. Limitations

One of the limitations of the study is not having controlled different physiological variables directly, by means of, for example, hormone controls. Another limitation is not having carried out a longitudinal study to observe the evolution of the athletes’ perception. These two aspects, on the other hand, are difficult to carry out due to the great difficulty of accessing a large sample of elite athletes. 

## 6. Practical Applications

Considering the results obtained, which reflect, on one hand, adaptive problems in athletes and, on the other, no appropriate level of institutional support against COVID-19, it would be advisable to develop targeted institutional programs to give psychological support to Olympic and Paralympic athletes in periods of intense social stress, such as pandemics or situations of confinement.

## 7. Conclusions

The situation caused by COVID-19 has had significant negative effects on the behavior, perception of threat, stress and training patterns of Olympic and Paralympic athletes who were candidates for the 2020 Tokyo Olympics. These athletes perceived how the pandemic negatively affected their workouts and their best athletic performance, even if it did not affect the recovery of their performance when the pandemic ended. The athletes of collective sports disrupted their training routine and their athletic performance more significantly than athletes of individual sports. The Paralympic felt more able to cope with their personal problems during the confinement than Olympic athletes and the athletes in individual sports felt more capable of coping with personal problems than athletes in collective sports. Only slightly more than half of the athletes received guidance from the Sports Federation. The female compared to male group were more affected psychologically in numerous variables and had their training routine disrupted by COVID-19 more than men. These data could be used to guide future interventions from sports authorities to improve the quality of life of athletes during major social crises.

## Figures and Tables

**Table 1 ijerph-18-12780-t001:** Age and anthropometric dimensions, training patterns, individual perceptions about COVID-19 crisis and psychological profile (perception of threat and stress).

	N	M	SD
Age	511	26.28	7.94
Weight (Kg)	495	67.86	13.83
Height (cm)	503	172.92	10.28
Training routine disrupted by COVID-19	496	3.24	0.84
Athletic performance affected by the confinement	501	3.05	0.84
Hours per week of training before COVID-19	493	17.30	9.53
Hours per week of training during COVID-19	494	8.44	7.16
Gravity of the pandemic	510	7.87	1.73
Stressed by the confinement	501	2.57	0.89
Worried about reducing their athletic capability	502	2.88	0.88
Worried about not being able to participate in competitions	502	3.04	0.93
Worried about their financial situation	499	2.49	1.07
Feeling lonely during confinement	502	1.78	0.87
Missed interacting with other athletes during the confinement	502	3.08	0.86
Impact of the pandemic on oneself	510	6.98	1.99
Estimation of the duration of the pandemic	511	6.34	1.60
Symptoms of COVID-19	509	1.44	1.18
Concern about the pandemic	511	6.75	2.09
Emotional affectation by the pandemic	511	5.69	2.40
Affected by unexpected event	504	2.82	1.01
Lack of control over important events	503	2.50	1.15
Nervous/anxious or stressed	505	2.90	1.08
Able to cope with personal problems	500	3.78	1.04
Feel that the events are going well	502	3.63	0.87
Incapable of facing their responsibilities	500	2.38	1.03
Able to manage their problems	500	3.84	0.91
Feel in control of all aspects	501	3.05	1.10
Angry by events that were out of their control	501	2.67	1.05
Feel the problems piling up	498	1.98	0.99

**Table 2 ijerph-18-12780-t002:** Perception of threat, stress and training patterns about COVID-19 crisis.

IV	Comparisons	N	N = Yes	%	N = No	%	X^2^	*p*
Have you been tested for COVID-19?								
Sport specialty	Individual	511	224	68.3%	104	31.7%	7.20	0.027
	Collective		107	58.5%	76	41.5%		
Have you been infected by COVID-19?								
Gender	Male	511	11	4.3%	246	95.7%	5.34	0.021
	Female		24	9.4%	230	90.6%		
Did you feel more tired than usual during the confinement?								
Gender	Male	511	106	41.2%	151	58.8%	11.58	0.001
	Female		143	56.3%	111	43.7%		
Did you have difficulty sleeping during the confinement?								
Gender	Male	511	105	40.9%	152	59.1%	10.41	0.001
	Female		140	55.1%	114	44.9%		
Did you increase your use of tobacco during the confinement?								
Gender	Male	504	5	2%	249	98%	5.37	0.020
	Female		15	6%	235	94%		
Did you increase your alcoholic intake during the confinement?								
Age	25 or Less	511	17	5.9%	269	94.1%	10.85	0.001
	26 or More		33	14.7%	192	85.3%		
Did you take psychotropic medications for mental health during the confinement (for example, for anxiety, depression, panic)?								
Age	25 or Less	505	2	0.7%	280	99.3%	5.31	0.021
	26 or More		8	3.6%	215	96.4%		
Did you receive guidance from your sports organization about COVID-19?								
Age	25 or Less	496	131	47.3%	146	52.7%	5.13	0.023
	26 or More		126	57.5%	93	42.5%		
Participation	Olympic	496	213	49.1%	221	50.9%	10.41	0.001
	Paralympic		44	71%	18	29%		
Did you receive guidance from your coach about COVID-19?								
Gender	Male	511	153	60.2%	101	39.8%	9.59	0.002
	Female		188	73.2%	69	26.8%		
Sport specialty	Individual	511	206	62.8%	122	37.2%	6.36	0.012
	Collective		135	73.8%	48	26.2%		
Did your personal best athletic performance diminish due to the COVID-19 confinement?								
Age	25 or Less	504	231	82.2%	50	17.8%	6.05	0.014
	26 or More		163	73.1%	60	26.9%		
Academic training	High school	504	126	78.3%	35	21.7%	6.85	0.032
	Professional		84	87.5%	12	12.5%		
	University		184	74.5%	63	25.5%		

**Table 3 ijerph-18-12780-t003:** Correlational analysis of perception of threat and stress variables.

	1	2	3	4	5	6	7	8	9	10	11	12	13	14	15
1. Impact of the pandemic on oneself	1														
2. Estimation of the duration of the pandemic	0.27 **	1													
3. Symptoms of COVID	0.07	0.11 *	1												
4. Concern about the pandemic	0.40 **	0.34 **	0.12 **	1											
5. Emotional affectation by the pandemic	0.40 **	0.23 **	0.13 **	0.51 **	1										
6. Affected by unexpected event	0.3 4**	0.10 *	0.12 **	0.30 **	0.45 **	1									
7. Lack of control over important events	0.26 **	0.07	0.10 *	0.2 3**	0.43 **	0.53 **	1								
8. Nervous/anxious or stressed	0.23 **	0.18 **	0.11 *	0.30 **	0.50 **	0.48 **	0.54 **	1							
9. Able to cope with personal problems	−0.05	−0.06	−0.10 *	−0.13 **	−0.23 **	−0.23 **	−0.31 **	−0.30 **	1						
10. Feel that the events are going well	−0.14 **	−0.13 **	−0.12 **	−0.15 **	−0.37 **	−0.29 **	−0.35 **	−0.34 **	−0.45 **	1					
11. Incapable of facing their responsibilities	0.14 **	0.08	0.15 **	0.14 **	0.32 **	0.40 **	0.44 **	0.47 **	−0.32 **	−0.35 **	1				
12. Able to manage their problems	−0.06	−0.02	−0.08	−0.06	−0.21 **	−0.23 **	−0.36 **	−0.30 **	0.52 **	0.50 **	−0.35 **	1			
13. Feel in control of all aspects	−0.14 **	−0.04	−0.03	−0.12 **	−0.33 **	−0.33 **	−0.40 **	−0.33 **	0.40 **	0.47 **	−0.35 **	0.46 **	1		
14. Angry by events that were out of control	0.23 **	0.11 *	0.08	0.20 **	0.41 **	0.43 **	0.44 **	0.45 **	−0.26 **	−0.38 **	−0.42 **	−0.23 **	−0.33 **	1	
15. Feel the problems piling up	0.14 **	0.16 **	0.14 **	0.22 **	0.40 **	0.44 **	0.49 **	0.53 **	−0.36 **	−0.43 **	0.48 **	−0.39 **	−0.35 **	0.55 **	1
M	6.98	6.34	1.44	6.75	5.69	2.82	2.50	2.90	3.78	3.63	2.38	3.84	3.05	2.67	1.98
SD	1.99	1.60	1.18	2.09	2.40	1.01	1.15	1.08	1.04	0.87	1.03	0.91	1.10	1.05	0.99
N	510	511	509	511	511	504	503	505	500	502	500	500	501	501	498

* *p* ≤ 0.05; ** *p* ≤ 0.01.

## Data Availability

Data will be available upon reasonable request to the corresponding author.

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
