# Peer review of "Effects of COVID-19 Home Confinement on Behavior, Perception of Threat, Stress and Training Patterns of Olympic and Paralympic Athletes"

_ijerph, 2021, doi:10.3390/ijerph182312780_

Round 1
Reviewer 1 Report
Thank you for your submitted manuscript entitled, “Effects of COVID-19 home confinement on behaviour, perception of threat, stress and training patterns of Olympic and Paralympic athletes”. The authors have conducted a study with an aim that is commendable, considering the situation generated throughout society by COVID-19 and that has greatly affected athletes, especially the largest sporting event worldwide, the Olympic Games, postponing the 2020 Tokyo Olympics.
Manuscript is well written and clearly justifies the importance of novelty of the study, describes the procedures well, and places the findings in context of previous work. There are few studies that provide insight, asked directly to athletes through surveys, into the Effects of COVID-19 home confinement in Olympic and Paralympic athletes, involving great difficulty in achieving such a large sample (447 Olympic and 64 Paralympic) that it can be very representative to know behaviour, perception of threat, stress and training patterns of this group.
It is an interesting area of study and I think that the objectives are well described and that they are suitable. The data used in the study are relevant and analyzed rigorously. The three tables correctly synthesize the main results and the two appendices complete interesting information to be taken into consideration by the researchers. The outcomes of the study are very likely to be valuable to Institutions such as the International Olympic Committee, Sports Federations, coaches, players and scientists working in the sport performance area.
This is a very good manuscript that should be acceptable with minor revision to address the following comments:
P1 L42-43: In the sentence “On March 11, 2020 the new coronavirus disease 2019 (COVID-19) caused by SARS-CoV-2 was declared a pandemic ...” it is not necessary to rewrite “SARS-CoV- 2” already seems very repetitive as you have already written it before on line 41, so, for example, you can write“ On March 11, 2020 this coronavirus disease was declared a pandemic… ”
P2 L49: Better to write “from 2 to 22 September” instead of “from September 2 to September 22”
P2 L79: I think there is an error after “COVID-19 confinement ·, since there should be “. ” instead of ",".
P3 L96: The data (N = 511), with 447 Olympic and 64 Paralympic athletes is unbalanced, with many more Olympic athletes. Although I have verified through information on the internet that the Spanish participation for 2020 Tokyo Olympics was 326 Olympic athletes compared to 142 Paralympic athletes, it would have been ideal to have a larger sample of Paralympic. I imagine that this decompensation will be justified for reasons that the authors know about difficulty in accessing the sample or possible difficulties in completing it depending on the disability. However, the sample size is good.
P4 L154: It is not necessary to write “Applying the Wilcoxon test”, since the different ones were previously explained as it has been previously explained in “2.3. Statistical analysis".
P4 L159: It is not necessary to write “U Mann-Whitney” since the different ones were previously explained as it has been previously explained in “2.3. Statistical analysis ".
P5 L175: It is not necessary to write “U Mann-Whitney” since the different ones were previously explained as it has been previously explained in “2.3. Statistical analysis ".
P5 L192: It is not necessary to write “during the confinement”
P5 L194: It is not necessary to write “Kruskal-Wallis” since the different ones were previously explained as it has been previously explained in “2.3. Statistical analysis ".
P5 L205: It is not necessary to write “during the confinement”
P5 L207: It is not necessary to write “U Mann-Whitney” since the different ones were previously explained as it has been previously explained in “2.3. Statistical analysis ".
P5 L210: It is not necessary to write “during the confinement”
P6 L217: It is not necessary to write “U Mann-Whitney” since the different ones were previously explained as it has been previously explained in “2.3. Statistical analysis ".
P6 L236: It is not necessary to write “U Mann-Whitney” since the different ones were previously explained as it has been previously explained in “2.3. Statistical analysis ".
P9 L325: In “4.5. Effect of sports categories ", if what I mentioned in “P3 L96: The data (N = 511), with 447 Olympic and 64 Paralympic athletes ... ” the authors, only in the case of a known or very or likely to explain the difference in sample sizes, this cause could be mentioned.
Author Response
Thank you for your submitted manuscript entitled, “Effects of COVID-19 home confinement on behaviour, perception of threat, stress and training patterns of Olympic and Paralympic athletes”. The authors have conducted a study with an aim that is commendable, considering the situation generated throughout society by COVID-19 and that has greatly affected athletes, especially the largest sporting event worldwide, the Olympic Games, postponing the 2020 Tokyo Olympics.
Manuscript is well written and clearly justifies the importance of novelty of the study, describes the procedures well, and places the findings in context of previous work. There are few studies that provide insight, asked directly to athletes through surveys, into the Effects of COVID-19 home confinement in Olympic and Paralympic athletes, involving great difficulty in achieving such a large sample (447 Olympic and 64 Paralympic) that it can be very representative to know behaviour, perception of threat, stress and training patterns of this group.
It is an interesting area of study and I think that the objectives are well described and that they are suitable. The data used in the study are relevant and analyzed rigorously. The three tables correctly synthesize the main results and the two appendices complete interesting information to be taken into consideration by the researchers. The outcomes of the study are very likely to be valuable to Institutions such as the International Olympic Committee, Sports Federations, coaches, players and scientists working in the sport performance area.
Thank you very much for all your positive feedback.
This is a very good manuscript that should be acceptable with minor revision to address the following comments:
P1 L42-43: In the sentence “On March 11, 2020 the new coronavirus disease 2019 (COVID-19) caused by SARS-CoV-2 was declared a pandemic ...” it is not necessary to rewrite “SARS-CoV- 2” already seems very repetitive as you have already written it before on line 41, so, for example, you can write“ On March 11, 2020 this coronavirus disease was declared a pandemic… ”
We have changed "SARS-CoV- 2" to "On March 11, 2020 this coronavirus disease was declared a pandemic ..."
P2 L49: Better to write “from 2 to 22 September” instead of “from September 2 to September 22”
We have made the proposed change.
P2 L79: I think there is an error after “COVID-19 confinement ·, since there should be “. ” instead of ",".
We have included "." Instead of ",".
P3 L96: The data (N = 511), with 447 Olympic and 64 Paralympic athletes is unbalanced, with many more Olympic athletes. Although I have verified through information on the internet that the Spanish participation for 2020 Tokyo Olympics was 326 Olympic athletes compared to 142 Paralympic athletes, it would have been ideal to have a larger sample of Paralympic. I imagine that this decompensation will be justified for reasons that the authors know about difficulty in accessing the sample or possible difficulties in completing it depending on the disability. However, the sample size is good.
Indeed, there is a certain decompensation between the sample of Olympic and 64 Paralympic athletes. We used the same strategies to access the two samples but got fewer responses. We believe that, in addition to depending on different institutions, probably some types of disability may have influenced the difficulty of completing the questionnaire. It would be interesting for future occasions to look for strategies to expand the number of responses.
P4 L154: It is not necessary to write “Applying the Wilcoxon test”, since the different ones were previously explained as it has been previously explained in “2.3. Statistical analysis".
We have suppressed "Applying the Wilcoxon test"
P4 L159: It is not necessary to write “U Mann-Whitney” since the different ones were previously explained as it has been previously explained in “2.3. Statistical analysis ".
We have suppressed "U Mann-Whitney"
P5 L175: It is not necessary to write “U Mann-Whitney” since the different ones were previously explained as it has been previously explained in “2.3. Statistical analysis ".
We have suppressed "U Mann-Whitney"
P5 L192: It is not necessary to write “during the confinement”
We have removed “during the confinement”
P5 L194: It is not necessary to write “Kruskal-Wallis” since the different ones were previously explained as it has been previously explained in “2.3. Statistical analysis ".
We have suppressed " Kruskal-Wallis"
P5 L205: It is not necessary to write “during the confinement”
We have removed “during the confinement”
P5 L207: It is not necessary to write “U Mann-Whitney” since the different ones were previously explained as it has been previously explained in “2.3. Statistical analysis ".
We have suppressed "U Mann-Whitney"
P5 L210: It is not necessary to write “during the confinement”
We have removed “during the confinement”
P6 L217: It is not necessary to write “U Mann-Whitney” since the different ones were previously explained as it has been previously explained in “2.3. Statistical analysis ".
We have suppressed "U Mann-Whitney"
P6 L236: It is not necessary to write “U Mann-Whitney” since the different ones were previously explained as it has been previously explained in “2.3. Statistical analysis ".
We have suppressed "U Mann-Whitney"
P9 L325: In “4.5. Effect of sports categories ", if what I mentioned in “P3 L96: The data (N = 511), with 447 Olympic and 64 Paralympic athletes ... ” the authors, only in the case of a known or very or likely to explain the difference in sample sizes, this cause could be mentioned.
As we explained previously, we are not aware of the reasons for this sample difference. For future studies, we will take into consideration the possible difficulties of some Paralympic athletes in completing questionnaires and other variables that may have affected these lower response rates compared to Olympic athletes.
Reviewer 2 Report
Effects of COVID-19 home confinement on behavior, perception of threat, stress and training patterns of Olympic and Paralympic athletes.
The paper presents a descriptive study exploring and comparing the psychological effects of lockdown on Olympic and Paralympic athletes’ behaviors, perception of threat, stress, state of mind and training pattern. The topic is relevant and has potential applied implications. However, there are some limitations -fairly easy to fix- that should be addressed before publication.
There are some typos to correct and some references in the text do not follow APA norms (e.g., “Tomczak & Tomczak”, p. 138).
Results
The choice of statistical strategies is well explained and adequate. Results are clearly presented.
However, items composing scales (like Perceived Stress, or Perceived Stress Scale in the present study) are usually not analyzed one by one but collapsed, aggregated into a single index (e.g., total scale) or subdimensions, and internal consistency indicators (e.g., Cronbach alpha) are provided. Such strategy allows increasing reliability of instruments and results.
I guess authors used the term modulated to refer to the effects of individual differences (like for example in Donaldson, Nakamura and Moinpour, 2009). However, gender and other socio demographic variables are usually considered and labelled as “moderator” in the literature. Authors should explain (with one sentence), their choice for the term “regulator” to avoid confusion.
Donaldson, G.W., Nakamura, Y. & Moinpour, C. (2009). Mediators, moderators, and modulators of causal effects in clinical trials––Dynamically Modified Outcomes (DYNAMO) in health-related quality of life. Quality of Life Research, 18, 137–145. https://doi.org/10.1007/s11136-008-9439-x
Author Response
Effects of COVID-19 home confinement on behavior, perception of threat, stress and training patterns of Olympic and Paralympic athletes.
The paper presents a descriptive study exploring and comparing the psychological effects of lockdown on Olympic and Paralympic athletes’ behaviors, perception of threat, stress, state of mind and training pattern. The topic is relevant and has potential applied implications. However, there are some limitations -fairly easy to fix- that should be addressed before publication.
Thank you very much for your positive feedback.
There are some typos to correct and some references in the text do not follow APA norms (e.g., “Tomczak & Tomczak”, p. 138).
Thank you very much for your precision on some typos to correct and some references in the text do not follow APA norms. We have revised the text and corrected some typographical errors and we have also corrected the reference to “Tomczak & Tomczak”, p. 138 and also another from Navon-Eyal & Taubman-Ben-Ari, p 10.Results
The choice of statistical strategies is well explained and adequate. Results are clearly presented.
However, items composing scales (like Perceived Stress, or Perceived Stress Scale in the present study) are usually not analyzed one by one but collapsed, aggregated into a single index (e.g., total scale) or subdimensions, and internal consistency indicators (e.g., Cronbach alpha) are provided. Such strategy allows increasing reliability of instruments and results.
Thank you very much, indeed, in the case of the Perceived Stress by Psychometric Properties of a European Spanish Version of the Per-ceived Stress Scale (PSS-10), as it is composed of a single factor, we have decided to carry out a more "qualitative" analysis, sacrificing internal consistency, to be able to observe nuances of the 10 items of the questionnaire as some of them are very interesting for the exceptional situation generated by COVID-19 and in this way to be able to discuss some interesting results and compare them with other studies that used categories in some similar cases to these items.
I guess authors used the term modulated to refer to the effects of individual differences (like for example in Donaldson, Nakamura and Moinpour, 2009). However, gender and other socio demographic variables are usually considered and labelled as “moderator” in the literature. Authors should explain (with one sentence), their choice for the term “regulator” to avoid confusion.
Donaldson, G.W., Nakamura, Y. & Moinpour, C. (2009). Mediators, moderators, and modulators of causal effects in clinical trials––Dynamically Modified Outcomes (DYNAMO) in health-related quality of life. Quality of Life Research, 18, 137–145. https://doi.org/10.1007/s11136-008-9439-x
Thank you very much for this important detail, we have changed the expression "modulated" to the broader expression "analyzed", to avoid confusion.